# Effects of Resistance Training on C-Reactive Protein and Inflammatory Cytokines in Elderly Adults: A Systematic Review and Meta-Analysis of Randomized Controlled Trials

**DOI:** 10.3390/ijerph19063434

**Published:** 2022-03-14

**Authors:** Sang-Dol Kim, Young-Ran Yeun

**Affiliations:** Department of Nursing, College of Health Science, Kangwon National University, Samcheok 25949, Korea; srksd1965@gmail.com

**Keywords:** resistance training, CRP, cytokines, inflammation, elderly

## Abstract

Chronic low-grade inflammation that accompanies aging is associated with adverse health outcomes and may exacerbate the severity of infectious disease such as COVID-19. Resistance training (RT) has the potential to improve chronic low-grade inflammation, but the evidence remains inconclusive. This study evaluated the effects of RT on chronic low-grade inflammation in elderly adults. MEDLINE, EMBASE, Cochrane Library, CINAHL, RISS, NDSL, and KoreaMed were searched. We included studies that assessed the effect of RT on C-reactive protein (CRP), interleukin (IL)-6, IL-10, and tumor necrosis factor (TNF)-α in those aged ≥60 years. The effect size was estimated using fixed or random-effects models. Subgroup analysis was performed regarding age, health status, training method, number of exercises, intensity, weekly frequency, and duration. In the 18 randomized controlled trials (539 patients) included, RT was effective in alleviating CRP (effect size = −0.72, 95% confidence interval = −1.06 to −0.38, *p* < 0.001), IL-10 (−3.34, −6.16 to −0.53, *p* = 0.02), and TNF-α (−0.56, −1.08 to −0.03, *p* = 0.04) in elderly adults and tended to reduce IL-6 (−0.59, −1.18 to 0.00, *p* = 0.05). Subgroup analyses showed CRP reduction regardless of age, training method, number of exercises, intensity, weekly frequency, and duration. RT can be used to ameliorate chronic low-grade inflammation in elderly adults.

## 1. Introduction

The global coronavirus disease 2019 (COVID-19) pandemic caused by severe acute respiratory syndrome virus (SARS-CoV-2) is still ongoing. As of 24 November 2021, the cumulative number of confirmed cases worldwide was 257,046,038, and the death toll was 5,154,865 [1]. In particular, people over 60 years of age are considered a major high-risk group for COVID-19. Data from 16 countries, reporting a relatively high number of COVID-19 cases, showed that the death rate from COVID-19 among individuals aged ≥65 years was 62.1 times higher than among those aged <54 years [2]. In South Korea, approximately 21% of COVID-19 cases involve individuals aged >60 years, accounting for 92% of deaths [1]. 

A normal inflammatory response is characterized by a transient increase in inflammatory activity that develops when an infection is present and resolves once the infection has subsided [3]. However, certain social, psychological, environmental, and biological factors have been associated with inhibition of acute inflammation resolution and development of a low-grade, non-infectious chronic inflammation state [4]. Both short- and long-term changes in the inflammatory response can affect immunity and cause major changes in tissues and organs, as well as the normal cellular metabolism; this, in turn, may increase the risk of various communicable and noncommunicable diseases [5,6]. Chronic low-grade inflammation, also known as inflammaging, is associated with aging and is characterized by increases in serum inflammatory mediators such as C-reactive protein (CRP), interleukin (IL)-6, and tumor necrosis factor (TNF)-α [7]. 

Although the causes of chronic low-grade inflammation are not fully understood, results from previous studies have suggested that a combination of endogenous and non-endogenous social, environmental, and lifestyle risk factors may be involved [8]. Among endogenous causes are DNA damage, dysfunctional telomeres, and oxidative stress [9]. Non-endogenous contributors include chronic infection, lifestyle-induced obesity, unbalanced diet, and environmental toxicants [10,11,12,13]. Inflammation is linked to a decreased ability to mount effective immune responses in the elderly [14,15]. A decline in immune function not only increases the morbidity and mortality due to COVID-19, it also causes several degenerative diseases. Cancer, cardiovascular diseases, Alzheimer’s disease, Parkinson’s disease, osteoporosis, muscle loss, and depression are all related to inflammation [16,17,18].

Exercise and weight control have been suggested as methods for mitigating the negative effects of chronic inflammation. Obese people have continuously high levels of inflammation because fat cells secrete numerous types of inflammatory substances called adipokines, such as leptin, adiponectin, and resistin [19]. Proper exercise has a dual effect in reducing chronic inflammation via weight reduction and lowering adipokines in cells [20]. Although aerobic exercise is generally recommended, it is difficult for the elderly to perform weight-bearing exercises such as walking due to various problems including degenerative arthritis, cardiovascular disease, and diabetes [21]. In addition, the elderly have reduced muscle strength and range of motion, and the stimulation response period and nerve conduction speed are slow; thus, injuries and safety accidents easily occur during exercise [22]. Resistance training (RT), a strength training exercise involving progressive overload in which the muscles exert force against an external load, could be a safe and effective method of improving chronic low-grade inflammation in older individuals [23,24].

A previous study [25] reported that RT was associated with anti-inflammatory effects by decreasing serum levels of IL-6 and CRP, in addition to inducing changes in TNF-α gene expression in elderly women. However, another study [26] suggested that RT was not related to TNF-α, IL-6, IL-10, and CRP improvement. As a result, an integrated and clear conclusion on the effects of RT in the elderly is currently unavailable. The purpose of this review was to critically examine the effects of RT on chronic low-grade inflammation in elderly adults through a systematic review and meta-analysis of randomized controlled trials (RCTs). These findings will enable practitioners in prevention and treatment to present fundamental evidence for developing interventions to manage chronic low-grade inflammation.

## 2. Materials and Methods

### 2.1. Eligibility Criteria

The systematic review was conducted in accordance with the Preferred Reporting Items for Systematic reviews and Meta-Analyses (PRISMA) statement. Data selection was performed according to PICOTS-SD (Participants, Intervention, Comparisons, Outcomes, Timing of Outcome Measurement, Settings, Study Design), which is a guideline for conducting systematic reviews and meta-analyses. The participants (P) were elderly persons aged ≥60 years. The intervention (I) was RT. The comparison (C) was no intervention or other comparative intervention. The outcomes (O) were CRP, IL-6, IL-10, or TNF-α levels. The timing of outcome measurement (T) was post-intervention. The setting (S) was hospital, nursing home, or community, and the study design (D) was RCT. The exclusion criteria for the studies in this review were participant age <60 years; absence of RT or combined intervention (RT + other intervention); use of outcome variables other than CRP, IL-6, IL-10, or TNF-α, non-RCT study design; and unavailable mean and standard deviation values.

### 2.2. Literature Search

The searched databases included MEDLINE, EMBASE, Cochrane Library, CINAHL, RISS, NDSL, and KoreaMed. The search formula was combined by merging terms representing the elderly, RT, inflammatory biomarkers, and RCT. The search terms were as follows: (elder * OR old * OR aged OR aging OR senior) AND (resistance training OR strength training OR weight-lifting strengthening program OR weight-bearing strengthening program OR elastic OR machines OR equipment OR fitness) AND (biomarker * OR inflamm * OR marker * OR cytokine * OR C-reactive protein OR interleukin OR tumor necrosis factor) AND (randomized controlled trial OR controlled clinical trial OR randomized OR randomly). The search was conducted from the year of the first RT article to November 2021 and was limited to the English and Korean languages. 

### 2.3. Data Extraction 

First, the authors deleted duplicate articles and screened the titles and abstracts of the selected articles to eliminate irrelevant studies. Thereafter, the full texts of the remaining references were evaluated, and ineligible studies were excluded based on the criteria, and the reasons for exclusion were recorded. Two authors performed this process independently, and any disagreements were discussed until a compromise was achieved. The data of the finally selected studies were collected using the coding table. The coding table consisted of author, publication year, country of study, study subject, intervention method, control group, statistical value, methodological quality, etc.

### 2.4. Data Analysis 

The effect size and homogeneity of RT was analyzed using RevMan 5.3 (The Cochrane Collaboration, Oxford, UK). An effect size of 0.20–0.50 indicated a ‘small’ effect, 0.50–0.80 indicated a ‘medium’ effect, and ≥0.8 indicated a ‘large’ effect [27]. The statistical significance of the effect size was determined using the overall effect test and 95% confidence interval (CI) with the significance level set at 0.05 (*p* < 0.05). Heterogeneity was explored using *I*^2^ statistics, with values of 25%, 50%, and 75% considered low, moderate, and high heterogeneity, respectively [28]. When calculating the pooled effect size, the effect size was estimated using the fixed-effects model if the studies were homogeneous and with the random-effects model if the studies were heterogeneous. Publication bias was inspected with funnel plots. Subgroup analysis was performed by participant age, health status, training type, number of exercises, intensity, weekly frequency, and duration to identify potential moderators. According to the American College of Sports Medicine, moderate RT loading is defined as 60% of 1 repetition maximum (RM) [29]. Thus, intensity was classified as vigorous if the maximum intensity was >60 of I RM, and moderate if it was ≤60 of I RM.

### 2.5. Methodological Quality

Methodological quality was assessed using the Cochrane assessment of risk of bias (ROB). The tool is structured to evaluate the risk of bias in six domains as follows: random sequence generation, allocation concealment, blinding of participants and personnel, blinding of outcome assessment, incomplete outcome data, and selective outcome reporting. The risk of bias was determined as ‘low’, ‘high’, and ‘unclear’ for each item. The authors independently evaluated the items, and inconsistent items were reviewed and reevaluated.

## 3. Results

### 3.1. Literature Search

A total of 1044 articles were identified through the database search. After removing duplicate studies, the titles and abstracts of 546 studies suggested that 74 studies were eligible for inclusion. Thereafter, full text reviews identified 18 studies that met the inclusion criteria (Figure 1) [26,30,31,32,33,34,35,36,37,38,39,40,41,42,43,44,45,46]. 

### 3.2. Study Characteristics

The characteristics of the included studies are summarized in Table 1. The total number of participants was 539, with 268 in the intervention groups and 271 in the control groups. Sample sizes ranged from 14 to 48, with a median size of 30. The participants’ mean age ranged from 62.0 to 82.7 years, with an average of 70.2 years (76.7% women). Ten of the 18 interventions included healthy older individuals and 8 included individuals with specific diseases such as metabolic syndrome, type 2 diabetes, and cognitive impairment. Intervention durations ranged from 6 to 32 weeks, with a mean of 12 weeks. Of the 18 RT programs, 9 involved elastic band exercise, and the RT session frequency ranged from two to four sessions per week. The number of different muscle group exercises ranged from 6 to 14, and RT session duration ranged from 20 to 70 min (mode 60 min). Exercise intensity was moderate in 12 studies.

### 3.3. RT Effects

The results of the pooled meta-analysis for all outcomes are shown in Figure 2, and the results of the subgroup analysis are displayed in Table 2. Sixteen studies performed RT for CRP improvement, with a medium effect size of −0.72 (95% CI = −1.06 to −0.38, *p* < 0.001) and moderate heterogeneity (*I*^2^ = 69%). In the subgroup analysis, RCTs that included older individuals (>70 years), healthier individuals, machines or equipment, lower number of exercises (≤8), vigorous intensity, lower weekly frequency (≤2 times/week), and shorter duration (≤8 weeks) reported larger effect sizes for CRP. There was no publication bias on the Funnel plot.

Eleven studies examined IL-6, and a non-significant effect size was noted (−0.59, 95% CI = −1.18 to 0.00, *p* = 0.05, *I*^2^ = 84%) but tended towards reduction. In the subgroup analysis, IL-6 was significantly decreased in the studies that included healthier individuals, machines or equipment, lower number of exercises (≤8), and shorter duration (≤8 weeks). Publication bias was noted for IL-6 studies.

Three studies examined IL-10, with a large effect size of −3.34 (95% CI = −6.16 to −0.53, *p* = 0.002) and low heterogeneity (*I*^2^ = 45%). Unfortunately, subgroup analysis was not possible for IL-10 due to its heterogeneity between studies.

The effect on TNF-α was assessed in 11 studies, with a significant effect size of −0.56 (95% CI = −1.08 to −0.03, *p* < 0.001, *I*^2^ = 81%). RCTs with specific health conditions, machines or equipment, higher weekly frequency (>2 times/week), and longer duration (>8 weeks) showed significant decreases in the subgroup analysis. There was some publication bias.

### 3.4. Methodological Quality

In sequence generation, 11 studies were rated as unclear because randomization procedures were not sufficiently described. Four of the studies were classified as having low bias in relation to the allocation concealment domain. Only two studies had a low risk of bias regarding the blinding of participants and personnel. In the blinding of outcome assessment, all the studies were evaluated as having a low risk of bias because the RT results were calculated via blood analysis performed at an external clinical pathology center. Thirteen studies were determined as having low bias related to incomplete outcome domains (Figure 3).

## 4. Discussion 

As the prolonged COVID-19 pandemic has had a significant negative impact on the health of the elderly, public health interventions are needed to mitigate the situation. Chronic low-grade inflammation is associated with several adverse health outcomes and may exacerbate the severity and mortality of COVID-19. Therefore, this study was conducted to objectively synthesize studies evaluating the effects of RT on CRP and inflammatory cytokines in the elderly.

A meta-analysis of 18 RCTs demonstrated that RT had a medium effect size for reducing CRP, IL-10, and TNF-α in the elderly, and tended to reduce IL-6. Similarly, Sardel i et al. [47] showed that RT had a medium effect size in lowering CRP and tended to lower IL-6, but did not induce changes in TNF-α based on a meta-analysis of 13 RCTs in individuals > 50 years of age. Furthermore, Rose et al. [48] conducted a meta-analysis of 59 studies involving people of various ages and found that while RT had a small effect size for lowering CRP, it had no effect on IL-6 or TNF. Combining the above studies, CRP is reduced by RT regardless of age in the elderly, but IL-6, IL-10, and TNF-α may be affected by moderators such as participant age, participant condition, and intervention type.

CRP is a major marker of systemic inflammation. When inflammation or tissue damage occurs in the body, IL-6 stimulates hepatocytes to promote CRP production, raising the CRP concentration in the blood [49]. In addition to being a simple inflammatory marker, CRP is also associated with a decrease in muscle strength that causes deterioration in physical function among the elderly. According to a study by Schaap et al. [50], an increase in CRP and IL-6 levels by two to three times was associated with a decrease in grip strength by more than 40% in a sample of elderly patients. Another study showed a similar association between CRP and thigh muscle strength, with higher levels being associated with decreased thigh muscle strength [51]. In a rat study by Goodman et al., the authors found that administration of IL-6 and TNF-α was associated with muscle tissue destruction [52]. In a subgroup analysis of the current study, CRP levels showed statistically significant results, regardless of the training method, in those aged ≤70 and >70 years. Similarly, a previous meta-analysis investigating the effect of exercise on CRP levels found that exercising reduced CRP levels regardless of age or sex [53]. These results show that elderly individuals with a relatively high risk of injury from exercise can manage chronic inflammation using available RT. In addition, the effect size of CRP differed depending on exercise intensity, with the vigorous group having a larger effect size than the moderate group. Rose et al., who investigated the chronic inflammatory response according to the intensity of aerobic and resistance exercise, reported that higher-intensity training may be more efficacious than lower-intensity training for middle-aged adults, which supports the results of our study [54].

The pooled effect size for IL-6 was not significant for RT, but in the subgroup analysis, the effect size was significantly decreased in the studies that included healthier individuals, machines or equipment; lower number of exercises (≤8); and shorter duration (≤8 weeks). The pooled effect size for TNF-α was significantly decreased by RT. Particularly, it was significantly decreased in studies that consisted of healthier individuals, machines or equipment,; higher weekly frequency (>2 times/week); and longer duration (>8 weeks) in the subgroup analysis. Pérez Chaparro et al. [55] found that both RT alone and RT combined with aerobic exercise decreased IL-6 levels in HIV patients. Similarly, Hayashino et al. [56] discovered that aerobic and resistance exercises were more helpful for IL-6 in patients with type 2 diabetes when performed over a longer duration and in a larger number of sessions. Furthermore, Khalafi et al. [57] reported that exercise training significantly reduced IL-6 and TNF-α in postmenopausal women in a meta-analysis of 22 studies. Generally, TNF-α is produced in adipose tissues. It has been shown to aggravate inflammation and promote IL-6 secretion, which in turn, leads to metabolic disease [58,59]. Considering that TNF-α and IL-6 are closely related to an increase in adipose tissue [60] and a loss in muscle strength [61], the reduction of TNF-α and IL-6 may be because of a decrease in fat and an increase in muscle mass due to RT. In particular, IL-6 is secreted from both adipocytes and muscle cells. IL-6 secreted from adipocytes causes inflammatory action, whereas IL-6 secreted from muscle has anti-inflammatory action [62]. It is thought that the immune function of the elderly improves by exercising the muscles. Previous studies also emphasized the importance of increasing muscle mass in the anti-inflammatory effect of RT [26]. 

IL-10, while a pleiotropic cytokine, is an immune cytokine with anti-inflammatory potential. Petersen and Pedersen [63] reported that the process of muscle contraction during exercise may increase the transcription of IL-6 mRNA, resulting in activation of anti-inflammatory cytokines such as IL-10. Surprisingly, RT was found to lower IL-10 levels in this study, both in the elderly group (>70 years) and in the group with shorter duration (≤8 weeks), while previous studies showed no change or an increase in IL-10 after exercise. Gonzalo-Encabo et al. [64] showed that IL-10 increased after RT in adults with overweight and obesity, based on a systematic review of 27 studies. Yousefabadi et al. [65] in a meta-analysis of 20 RCTs also demonstrated that combined aerobic and resistance exercise increased IL-10 levels in patients with metabolic syndrome. 

Patients with pathological conditions, such as obesity, diabetes, and coronary artery disease often have high basal levels of pro-inflammatory cytokines [66]. As results, a reduction in IL-10 could be linked to a lower release. In addition, Cerqueira, et al. [67] reported that an elevation in pro-inflammatory cytokines was more evident after a prolonged bout of intense exercise. However, the intensity of RT was lower than that of the general population in this study in order to prevent inadvertent damage in elderly individuals. This could have resulted in a decrease in IL-10 levels. More research is needed to better understand the impact of RT on IL-10 in elderly adults without clinical comorbidities.

A strength of this study is that it is the first study to reveal the positive effects of RT on CRP and inflammatory cytokines in elderly adults aged ≥60 years through a systematic review and meta-analysis of RCTs. The limitations of this study in deriving such results are as follows. First, blinding of participants and personnel is an important aspect in evaluating the methodological quality of selected studies, but this was insufficiently performed in most studies. Second, a high level of heterogeneity was observed in the integrated meta-analysis, and a subgroup analysis was performed for further evaluation, but some subgroups still showed high heterogeneity. Therefore, the results of the meta-analysis should be interpreted with caution. Third, although a comprehensive method was applied for the literature search, publication bias was found. It is necessary to estimate the change in the pooled effect size after adjusting for publication bias.

## 5. Conclusions

The pooled results of the present study indicated that RT had significant effects on reducing CRP, IL-10, and TNF-α in the elderly, and tended to reduce IL-6. Particularly, RT reduced CRP regardless of the age of the elderly, training method, number of exercises, intensity, weekly frequency, and duration. However, IL-6 and TNF-α were affected by various moderators. Based on the results of this study, it is recommended that optimal RT for the elderly aged ≥60 years consists of vigorous intensity, lower number of exercises (≤8), lower weekly frequency (≤2 times/week), and shorter duration (≤8) using machines or equipment. RT may be safer to administer and could be an effective option to improve chronic low-grade inflammation in elderly adults.

## Figures and Tables

**Figure 1 ijerph-19-03434-f001:**
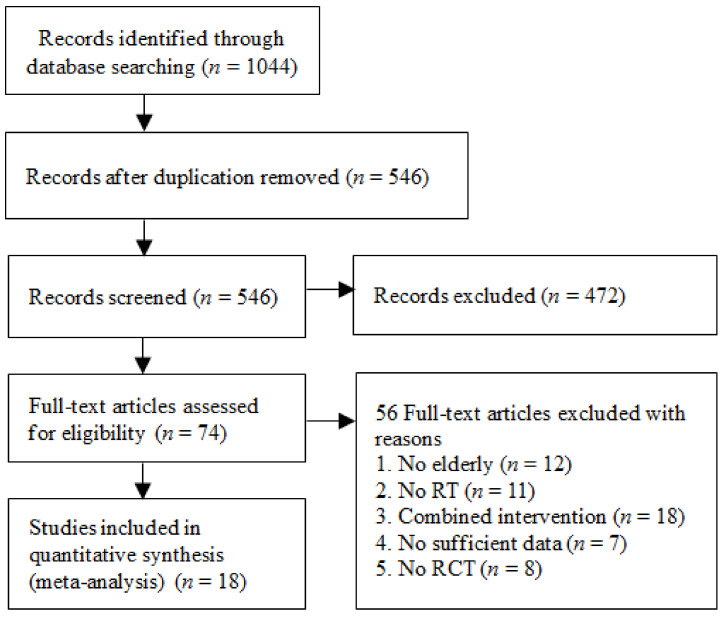
Flow diagram of the study selection process.

**Figure 2 ijerph-19-03434-f002:**
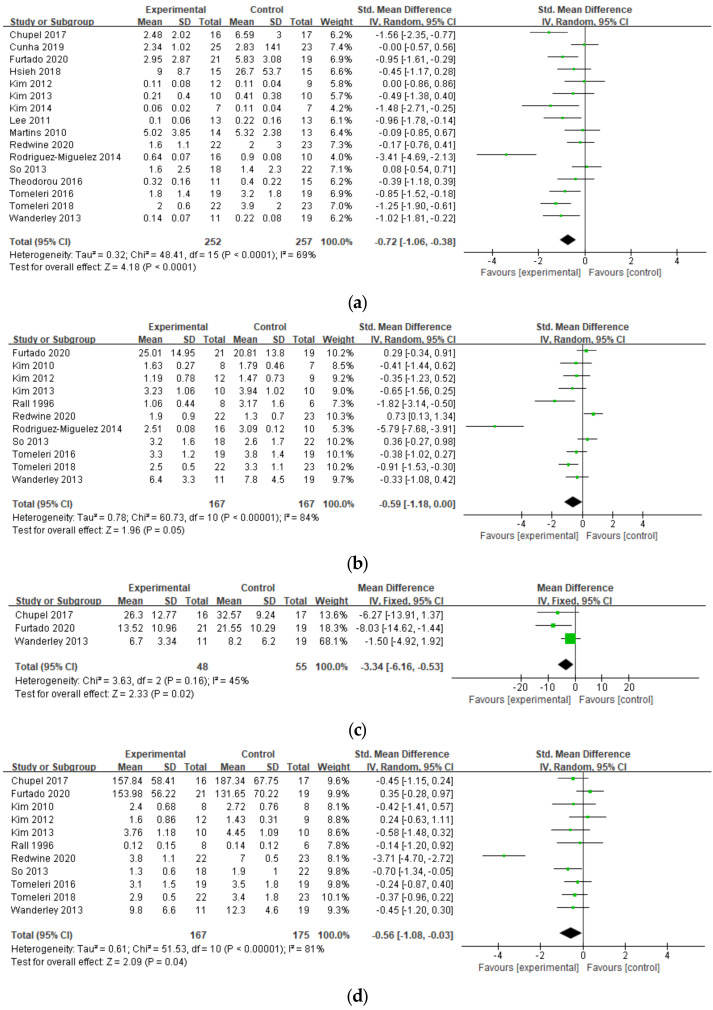
Forest plot of effect size by resistance training (RT) on (**a**) CRP, (**b**) IL-6, (**c**) IL-10, and (**d**) TNF-α.

**Figure 3 ijerph-19-03434-f003:**
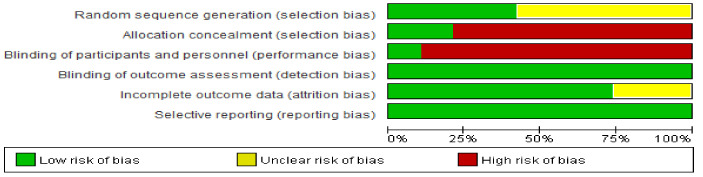
Methodological quality.

**Table 1 ijerph-19-03434-t001:** Characteristics of included studies.

First Author and Year	Participants: Health Condition, Sample Size *n* (EG, CG), Mean Age, % of Females	Interventions: Method, Number of Exercises (Ex), Intensity, Number of Sets and Repetitions, Minute, Weekly Frequency, Duration	Comparisons	Outcome Measures
Chupel 2017	Cognitive impairment, 33(16/17), 82.7, 100.0	Elastic band, 8–10 ex, vigorous, 1–2 × 10–12 Rep, 30–50 min, 2–3 times/week, 28 weeks	Usual care	CRP (↓), IL-10 (↔), TNF-α (↔)
Cunha 2019	Healthy 48(25/23), 70.3, 100.0	Machines, 8 ex, moderate, 1 × 10–15 Rep, 20 min, 3 times/week, 12 weeks	Inactive	CRP (↔)
Furtado 2020	Cognitive impairment 40(21/19), 80.5, 100.0	Elastic band, 8–10 ex, moderate, 2–3 × 10–15 Rep, 45 min, 2–3 times/week, 28 weeks	Usual care	CRP (↓), IL-6 (↔), IL-10 (↓), TNF-α (↔)
Hsieh 2018	Type 2 diabetes 30(15/15), 71.2, 63.3	Machines, 8 ex, vigorous, 3 × 8–12 Rep, NR, 3 times/week, 12 weeks	Usual care	CRP (↔)
Kim 2010	Healthy 16(8/8), 67.1, 100.0	Elastic band, 14 ex, moderate, 2 × 10–15 Rep, 60 min, 4 times/week, 12 weeks	Inactive	IL-6 (↔), TNF-α (↔)
Kim 2012	Type 2 diabetes 21(12/9), 68.8, 100.0	Machines, 8 ex, moderate, 2–3 × 10–15 Rep, 40–60 min, 3 times/week, 12 weeks	Usual care	CRP (↔), IL-6 (↔), TNF-α (↔)
Kim 2013	Healthy 20(10/10), 66.9, 100.0	Elastic band, 8 ex, vigorous, 3 × 10–12 Rep, 70 min, 3 times/week, 12 weeks	Inactive	CRP (↔), IL-6 (↔), TNF-α (↔)
Kim 2014	Healthy 14(7/7), 71.5, 100.0	Elastic band, 9 ex, moderate, 3 × 12 Rep, 60 min, 3 times/week, 12 weeks	Inactive	CRP (↓)
Lee 2011	Metabolic syndrome 26(13/13), 68.8, 100.0	Elastic band, 12 ex, moderate, 3 × 10–15 Rep, 50 min, 3 times/week, 12 weeks	Usual care	CRP (↓)
Martins 2010	Healthy 27(14/13), 73.2, 59.3	Callisthenic and elastic band, 8 ex, moderate, 6 × 8–12 Rep, 45 min, 3 times/week, 16 weeks	Inactive	CRP (↔)
Rall 1996	Healthy 14(8/6), 69.7, 31.4	Machines, 5 ex, vigorous, 3 × 8 Rep, 45 min, 2 times/week, 12 weeks	Inactive	IL-6 (↓), TNF-α (↔)
Redwine 2020	Heart Failure 45(22/23), 66.0, 13.5	Elastic band, NR, vigorous, NR, 60 min, 2 times/week, 16 weeks	Inactive	CRP (↔), IL-6 (↑), TNF-α (↓)
Rodriguez-Miguelez 2014	Healthy 26(16/10), 69.1, 73.1	Machines, 3 ex, vigorous, 3 × 8–12 Rep, NR, 2 times/week, 8 weeks	Inactive	CRP (↓), IL-6 (↓)
So 2013	Healthy 40(18/22), 69.8, 67.5	Elastic band, 14 ex, moderate, 2–3 × 15–25 Rep, 60 min, 3 times/week, 12 weeks	Inactive	CRP (↔), IL-6 (↔), TNF-α (↓)
Theodorou 2016	Coronary artery disease 26(11/15), 62.0, 0.0	Machines, 8 ex, moderate, 2 × 12–15 Rep, NR, 3 times/week, 32 weeks	Usual care	CRP (↔)
Tomeleri 2016	Obese 38(19/19), 68.2, 100.0	Machines, 8 ex, moderate, 3 × 10–15 Rep, 45–50 min, 3 times/week, 12 weeks	Inactive	CRP (↓), IL-6 (↔), TNF-α (↔)
Tomeleri 2018	Healthy 45(22/23), 70.4, 100.0	Machines, 8 ex, moderate, 3 × 10–15 Rep, NR, 3 times/week, 12 weeks	Inactive	CRP (↓), IL-6 (↓), TNF-α (↔)
Wanderley 2013	Healthy 30(11/19), 67.6, 73.3	Machines, 9 ex, moderate, 2 × 10–15 Rep, 50 min, 3 times/week, 32 weeks	Inactive	CRP (↓), IL-6 (↔), IL-10 (↔), TNF-α (↔)

EG, experimental group; CG, control group; NR, not reported; CRP, C- reactive protein; IL-6, interleukin-6; IL-10, interleukin-10; TNF-α, tumor necrosis factors –α; ↓, decreased; ↑, increased; ↔, no change.

**Table 2 ijerph-19-03434-t002:** Subgroup analysis of the effects of RT of elderly adults.

Subgroup	CRP				IL-6				TNF-α			
	Effect Size	95% CI	*n*	*I* ^2^	Effect Size	95% CI	*n*	*I* ^2^	Effect Size	95% CI	*n*	*I* ^2^
Age												
≤70 years	−0.69	−1.19, −0.20	9	73	−0.70	−1.43, 0.03	9	85	−0.73	−1.46, −0.00	8	84
>70 years	−0.83	−1.24, −0.42	7	54	−0.31	−1.49, 0.86	2	86	−0.15	−0.65, 0.34	3	46
Health condition												
Healthy	−0.10	−0.12, −0.07	8	87	−0.57	−0.65, −0.49	7	62	−0.47	−0.78, −0.16	6	36
Specific condition	−0.04	−0.08, 0.00	8	84	0.16	−0.18, 0.51	4	64	−0.73	−1.90, 0.45	5	92
Training method												
Elastic band	−0.64	−1.07, −0.21	8	61	0.17	−0.29, 0.63	5	50	−0.89	−1.87, 0.09	6	89
Machines	−0.82	−1.37, −0.27	8	77	−1.28	−2.21, −0.36	6	85	−0.33	−0.63, −0.04	5	0
Number of exercises												
≤8	−0.12	−0.15, −0.08	9	88	−0.59	−0.67, −0.51	6	27	−0.04	−0.17, 0.09	5	4
>8	−0.06	−0.09, −0.04	4	0	−0.20	−0.86, 0.46	3	53	−0.52	−0.93, −0.12	3	0
Intensity												
Moderate	−0.63	−0.93, −0.33	11	45	−0.23	−0.60, 0.14	7	46	−0.23	−0.52, 0.06	7	15
Vigorous	−1.11	−2.02, −0.20	5	84	−1.73	−3.84, 0.39	4	94	−1.21	−2.73, 0.31	4	91
Weekly frequency												
≤2	−0.26	−0.32, −0.20	2	0	−0.55	−0.63, −0.47	3	93	−1.93	−5.43, 1.57	2	96
>2	−0.05	−0.08, −0.03	12	69	−0.37	−0.61, −0.12	7	31	−0.38	−0.65, −0.10	7	0
Duration (week)												
≤8	−0.26	−0.32, −0.20	2	79	−0.59	−0.67, −0.50	3	62	−0.24	−0.87, 0.40	1	-
>8	−0.05	−0.08, −0.03	14	78	−0.14	−0.36, 0.09	8	68	−0.59	−1.18, −0.01	10	82

## Data Availability

Not applicable.

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
