# Peer review of "Effects of Resistance Training on C-Reactive Protein and Inflammatory Cytokines in Elderly Adults: A Systematic Review and Meta-Analysis of Randomized Controlled Trials"

_ijerph, 2022, doi:10.3390/ijerph19063434_

Round 1

Reviewer 1 Report

The purpose of this study was to examine the effects of resistance training on chronic low-grade inflammation in elderly adults through meta analyss (PRISMA) of many research articles. Generally, the importance of this investigation was recognized and well deserved.
The abstract covers strictly the main statements of the publication, Introduction is short but concise, Methods are adequate for the purposes.
However, the narration of introduction dealing with the specific mechanism about the resistance training (especially about the specific resistance mode) on chronic low-grade inflammation and inflammation factors in elderly adults with more references. 

There much be needs about reseach creativity in the introduction sections. 

It needs to be described about specific rationale and references in the section of methods

It might be needs of specific RT models and specific eligibility criteria (so much variables were in) in the methods with more references.

Discussion needs to be describes in more details (discuss about inflammatory factors and RT) with the other references. Moreover, there are too much grammatical errors which distract me to follow this manuscript.

Author Response

Dear reviewer, 

Thank you for your consideration.

Reviewer 2 Report

This interesting review and meta-analysis manuscript describes the effects of resistance exercise training on CRP and inflammatory cytokines in older adults. I commend the authors for undertaking this important work. I have a few main comments for the authors.

  1. In the introduction, Line 37-39, include that lifestyle behaviours have an influence on inflammation as well.
  2. Under 2.1, please include what the other comparative interventions were for used in this context.
  3. Under 2.4, was the significance level 0.5% which would equal p = 0.005? Or did the authors mean p ≤ 0.05? This was confusing.
  4. In Table 1, it would be helpful for the reader to see an ↑↓↔ symbol beside CRP and the cytokines to know what happened in each individual study.
  5. For the RT effects, if the p-value for IL-6 was 0.05 would this not be considered significant according to the criteria (if indeed p ≤ 0.05)?
  6. In Table 2, how was moderate and vigorous intensity resistance exercise defined?
  7. Line 190-192 - this sentence is very confusing and requires clarification.
  8. Line 194 - what type of exercise was included in this meta-analysis?
  9. Line 214 - do you mean adipose tissue not obese tissues?
  10. I believe the discussion could expand on the roles of CRP and the cytokines evaluated in the inflammatory process. For instance, IL-10, while a pleiotropic cytokine, is predominantly thought of as anti-inflammatory. Why would resistance training lead to a decrease in an anti-inflammatory cytokine?

Author Response

Dear Reviewer, 

Thank you for your consideration.

Reviewer 3 Report

The Review from Kim & Yeun is an interesting paper that summarizes the effects of resistance training on c-reactive protein and inflammatory cytokines in the elderly. Reviews with meta-analysis are very important in this field, to outline high evidence results. Nevertheless, the present version indicates significant methodological weakness. At first, the Authors should revise the main problems. After this, the Manuscript could be reviewed in more depth.

The Authors have missed listing the references of the original articles, which are focused on the review and the meta-analysis. Therefore, the reviewers or reads have no chance to comprehend and check them. I do not find the articles out of the first author and year of publishing!

However, I am familiar with the article of Despeghel et al. 2021. It is entitled “Effects of a 6 Week Low-Dose Combined Resistance and Endurance Training on T Cells and Systemic Inflammation in the Elderly”. In this article, combined resistance and endurance training took place. This is according to the authors’ methods (Line 76) an exclusion criterion! For this reason, the results of this article should exclude.

I would review the manuscript in detail if I could also get an overview of all the other original works and correction of the Paper of "Despeghel et al.".

Author Response

(The authors gave the same response as above.)

Round 2

Reviewer 2 Report

Thank you for addressing all my previous comments and queries. One minor change to make to the revised version: Line 269 - the word 'weaker' should be changed to 'lower'.

Author Response

Dear reviewer,

Thank you for your thoughtful suggestions and insights.

Reviewer 3 Report

Many thanks for the revision. There are no further comments on methodology and results.

I see the following points still to be worked on.

Introduction
The introduction describes the topic and gives important background knowledge. However, the rationale for writing this review is not entirely clear. After all, there are already two reviews that have dealt with the topic. These are also listed in the discussion (Rose et al. 2021/ Sardeli et al. 2018). Please clarify the contribution of your paper.

Discussion 
In the discussion, the results are detailed and discussed in the context of the existing literature. However, practical aspects should also be taken into account. Can recommendations for optimal strength training be derived? How should the intensity, volume, and duration of strength training be designed to provide the stated benefits?  These are important points to make the findings accessible to practitioners in prevention and therapy. 

Additional comments 
Line 59 as well as 61 - the term "substances" is very general. I would ask you to explain this a little more.

Line 204- Source

Author Response

(The authors gave the same response as above.)
